# The Effect of *Bacillus coagulans* Induced Interactions among Intestinal Bacteria, Metabolites, and Inflammatory Molecules in Improving Natural Skin Aging

Keiichi Hiramoto [1],*, Sayaka Kubo [2], Keiko Tsuji [2], Daijiro Sugiyama [2], Yasutaka Iizuka [2] and Hideo Hamano [2]

[1] Department of Pharmaceutical Sciences, Suzuka University of Medical Science, Suzuka 513-8670, Japan
[2] Research Department, Daiichi Sankyo Healthcare Co., Ltd., Tokyo 140-8170, Japan;
kubo.sayaka.hr@daiichisankyo-hc.co.jp (S.K.); tsuji.keiko.nj@daiichisankyo-hc.co.jp (K.T.);
sugiyama.daijiro.gz@daiichisankyo-hc.co.jp (D.S.); iizuka.yasutaka.r3@daiichisankyo-hc.co.jp (Y.I.);
hamano.hideo.gg@daiichisankyo-hc.co.jp (H.H.)
* Correspondence: hiramoto@suzuka-u.ac.jp; Tel.: +81-59-340-0575

**Abstract:** Background: Lactic acid bacteria consumption serves several health benefits to humans. However, their effect on natural skin aging is still unclear. Methods: This study examined the effects of skin naturalization (particularly skin drying) by administering a spore-bearing lactic acid bacteria (*Bacillus coagulans*) in mice for 2 years. Results: *B. coagulans* administration improved the natural skin of mice and significantly increased proportions of the genera *Bacteroides* and *Muribaculum*, among other intestinal bacteria. As metabolites, increases in nicotinic acid, putrescin, and pantothenic acid levels and a decrease in choline levels were observed. Increased hyaluronic acid, interleukin-10, and M2 macrophage levels indicate aging-related molecules in the skin. Intestinal permeability was also suppressed. Thus, these changes together improved natural skin aging. Conclusions: This study revealed that *B. coagulans* administration improved the natural skin aging in mice. This enhancement might be induced by the interaction of alterations in intestinal flora, metabolites, or inflammatory substances.

**Keywords:** *Bacillus coagulans*; hyaluronic acid; IL-10; nicotinic acid; pantothenic acid; intestinal permeability



## 1. Introduction

Skin aging is prominent on the face of the elderly, with prominent wrinkles, tawny skin tone, and irregular pigmentation, called blemishes. The skin then thickens and loses its elasticity. Furthermore, benign tumors and skin cancer may also occur. Such reactions are generally observed in exposed regions of the body, and the condition is termed photoaging because the cause thereof is exposure to ultraviolet rays [1–3]. In contrast, the unexposed skin becomes thin, lacks elasticity, and develops wrinkles, with negligible changes in color tone or stains, and the skin color lightens. These changes are termed endogenous aging, reflecting time-dependent changes due to factors within the body such as nutrition, genetics, metabolism, hormones, autonomic nerves, and immunity [1–4].

The functions of aging skin are also impaired, with decreased adaptability to environmental changes and a wide range of physiological functions such as cell regeneration, barrier function, chemical diffusion, endurance, wound healing, immune response, and DNA damage [1–5]. During natural aging, intrinsic aging weakens skin elasticity, causing sagging and uneven skin surfaces. In addition, the skin is dry, with a reduction in ceramides and natural moisturizing factors [6].

Furthermore, we have repeatedly investigated the effects of gender differences on natural aging [7,8]. A comparison of males and females showed that females suppressed aging more, and that estradiol was involved in this effect. It has been suggested that

estradiol inhibits the hyaluronic acid degrading enzyme and MMP-1, thereby preventing the decrease in hyaluronic acid and collagen in the skin and suppressing skin aging. In this way, since there are gender differences in the natural aging of the skin, gender differences were also investigated in this study.

The most remarkable factor in cellular senescence observed during natural aging is the presence of reactive oxygen species, which cause oxidative damage. When DNA damage cannot be completely repaired, it is removed by premature senescence or cell apoptosis. However, when the repair is defective and genetic abnormalities occur, cell function is altered. Oxidative stress also damages intracellular signaling pathways and alters gene activation, resulting in skin aging [9–11]. The skin plays an important role in protecting the body from the external environment. Therefore, skin aging has a great influence on physical health, and it is important to decelerate aging. In this study, we tested the applicability of lactic acid bacteria in improving natural aging.

Lactic acid bacteria are generally present in fermented foods and have several beneficial effects. They adjust the intestinal environment and regulate bowel movements [12], enhance the immune system [13], possess anti-allergic effects [14,15], and lower cholesterol [16]. Furthermore, lactic acid bacteria exhibit skin-beautifying effects and support the growth of *Staphylococcus epidermidis* on the skin surface [17]. *S. epidermidis* moisturizes the skin and improves skin texture; therefore, the ingestion of lactic acid bacteria is beneficial for the skin. In addition, lactic acid bacteria are effective against various types of damage to the skin induced by ultraviolet rays [18,19]. These effects of lactic acid bacteria are due to the enhancement of DNA repair function and the modulation of skin immunity. In summary, lactic acid bacteria may play an important role in alleviating the effects of photoaging [20,21].

Furthermore, we demonstrated that *B. coagulans* administration had an ameliorating effect on AOM+DSS-induced colon cancer [22]. It was revealed that *B. coagulans* induces an increase in transforming growth factor-β (TGF-β) and regulates the immune system. Chronic inflammation is thought to be one of the causes of aging, and inflammatory cytokines have been shown to play an important role [23]. *B. coagulans* was predicted to affect skin aging through its involvement with the immune system.

In addition, the ingestion of lactic acid bacteria affects and stabilizes the intestinal microflora [24], which subsequently have various effects on living organisms. An improvement in skin conditions has been reported with the administration of probiotics containing the genus *Lactobacillus*. For example, *Lactobacillus helveticus*, *L. brevis*, and *L. rhamnosus* reduce transepidermal water loss (TEWL) and improve skin dryness [25]. In addition, *L. johnsonii*, *L. reuteri*, and *L. plantarum* induce an increase in interleukin (IL)-10 and *L. rhamnosus* suppresses inflammation via mast cells [25]. In addition to *Lactobacillus*, the genus *Muribaculum* is involved in intestinal permeability [26] and the genus *Bacteroides* induces regulatory T cells [27]. Therefore, the administration of *Bacillus coagulans* may alter intestinal microflora and associated metabolites to control skin aging.

Although the effects of photoaging have been reported, the effect of lactic acid bacteria on natural aging is still unclear. This study aimed to examine the effects of spore-bearing lactic acid bacteria, which are resistant to gastric acid and actively reach the intestine, on natural skin aging.

## 2. Materials and Methods

### 2.1. Animal Experiments

Specific-pathogen-free (SPF) 8-week-old male and female hairless mice (Hos:HR-1: SLC, Hamamatsu, Shizuoka, Japan) were used. Mice were housed in an SPF room maintained at $23 \pm 1\ °C$ room temperature and $50 \pm 5\%$ humidity with a 12 h light/dark cycle to minimize stress. In addition, a fluorescent lamp with an ultraviolet filter (FLR110H, EX-D/M/36 WAN, Evryz Co., Ltd., Maebashi City, Gunma Prefecture, Japan) was used as the light source to block the influence of ultraviolet rays. Male and female mice were assigned to the following groups ($n$ = 5/group): a control group, a vehicle (distilled water)

group, and a *B. coagulans* administration group for each sex. The *B. coagulans* treated group was orally administered approximately 200,000 *B. coagulans* (Daiichi Sankyo) dissolved in distilled water three times per week for the duration of the two-year study. The daily dose of *B. coagulans* given to the mice was determined by converting it to the amount that a human weighing 60 kg would take once a day. On the last day of the study (one day after the last dose), samples (plasma, dorsal skin, and fresh feces) were collected. This study was approved by the Suzuka University of Medical Science Animal Experiment Ethics Committee on 25 September 2014 and performed in strict accordance with the recommendations of the Guide for the Care and Use of Laboratory Animals of the Suzuka University of Medical Science (Approval number: 34). All surgeries were performed on mice under pentobarbital anesthesia and efforts were made to minimize animal suffering.

### 2.2. Measurement of TEWL and Skin Hydration Levels

On the final day of the study, TEWL and skin hydration levels in the dorsal skin were measured using a previously described method [28]. TEWL, a permeability measurement that reflects skin barrier function, was measured using a Tewameter TM300 (Courage + Khazaki Electronic GmbH, Cologne, Germany). Skin hydration levels were assessed using a corneometer CM825 (Courage + Khazaki Electronic GmbH). Skin hydration was determined by measuring capacitance in arbitrary units. TEWL and hydration levels were measured at the same place on the back of each mouse under the same pressure in order to match the conditions. Furthermore, the room temperature was kept constant.

### 2.3. Evaluation of Wrinkles

Using the method reported by Bissett et al., wrinkles on the back of hairless mice were observed on the final day of the study and scored using the following scale: 0, no wrinkles; 1, light wrinkles; 2, slightly deep wrinkles; and 3, deep wrinkles [29].

### 2.4. Preparation and Staining of the Dorsal Skin Sections

Skin samples collected on the final day of the experiment were fixed with 4% phosphate-buffered paraformaldehyde and cryo-embedded in Tissue Tek O.C.T with a compound (Sakura Finetek, Tokyo, Japan). The skin was cut into cryosections to a thickness of 5 μm and air-dried. These sections were stained with hematoxylin and eosin, which are common in tissues. Using this HE-stained image, the thickness of the entire skin (from stratum corneum to dermis) was determined. Ten regions were randomly selected within the acquired image. Skin sections were then stained with toluidine blue. Toluidine blue can stain mast cells. Similarly, 10 skin images were randomly selected, and mast cells were quantified by counting the number of cells per square millimeter in that area under a microscope. In addition, Masson's Trichrome staining was performed using a Trichrome Modified Masson's Staining Kit (Scytek Laboratories, Logan, UT, USA) according to the manufacturer's instructions to observe skin collagen.

### 2.5. Measuring the Plasma Levels of Hyaluronic Acid, IL-6, Tumor Necrosis Factor-α, IL-10, and Skin Levels of Transforming Growth Factor-β, Hyaluronic Acid, Ionized Calcium-Binding Adapter Molecule 1, CC-Chemokine Receptor 7, and Cluster of Differentiation 163

The levels of hyaluronic acid, IL-6, tumor necrosis factor (TNF)-α, IL-10, transforming growth factor (TGF)-β, ionized calcium-binding adapter molecule 1 (Iba1) (a marker of macrophage), CC-chemokine receptor 7 (CCR7) (a marker of M1 type macrophage), and a cluster of differentiation (CD)163 (a marker of M2 type macrophage) were determined using commercial enzyme-linked immunosorbent assay kits (IL-6, TNF-α, IL-10, and TGF-β; R&D Systems, Minneapolis, MN, USA: Iba1 and CD163; MyBiosource, San Diego, CA, USA: CCR7; ELK Biotechnology, Denver, CO, USA) in accordance with the manufacturer's instructions.

### 2.6. Microbiome Analysis of Intestinal Flora

Fresh feces collected from each mouse on the final day of the study was subjected to 16S rRNA gene sequencing to reveal the bacterial phylogenetic composition. DNA extraction from stool specimens and polymerase chain reaction for 16S rRNA gene sequences were performed using the method described by Murakami et al. [30].

### 2.7. Stool Metabolome Analysis

To evaluate the effects of *B. coagulans* intake on metabolites, a comprehensive analysis of fecal metabolites was performed. Metabolites were extracted from stool samples, followed by an analysis of the extracted metabolites using capillary electrophoresis/time of flight mass spectrometry.

### 2.8. Statistical Analyses

All data are presented as mean ± standard deviation. The results were analyzed using the Microsoft Excel 2010 software (Microsoft Corporation, Redmond, WA, USA). Differences between groups were evaluated using one-way analysis of variance, followed by Tukey's post hoc test, using the SPSS version 20 software (SPSS Inc., Chicago, IL, USA). The results were considered statistically significant at $p$-values < 0.05 or <0.01. In addition, Python (version 3.7.3) was used for the analysis of intestinal bacteria and metabolites. Scipy (version 1.3.1) was used for statistical testing. Wilcoxon's rank sum test was performed for comparison between the control group and the *B. coagulans* group with $p$-values < 0.05 considered as statistically significant.

## 3. Results

### 3.1. Effects of B. coagulans Administration on TEWL, Skin Hydration, and Skin Conditions in Aging Mice

Water retention, an indicator of skin dryness, decreased with age in both male and female mice. Additionally, water transpiration increased with age in both male and female mice. This effect was significantly suppressed by long-term administration of *B. coagulans* (Figure 1A,B). Conversely, skin hydration (measured in terms of the corneometer value) decreased with age, but *B. coagulans* administration suppressed the decrease (Figure 1C,D). Skin thickness increased in both males and females, but this was suppressed by *B. coagulans* administration, with the effect being more pronounced in females than in males (Figure 1E–G). Moreover, after long-term administration of *B. coagulans*, dorsal skin wrinkles were improved. This effect was also higher in females than in males (Figure 1H–K). Histological examination of the skin of male and female mice showed increased collagen fiber content after long-term *B. coagulans* administration (Figure 1L,N).

### 3.2. Effects of B. coagulans Administration on the Levels of Hyaluronic Acid, IL-6, TNF-α, IL-10, and TGF-β in Aging Mice

Hyaluronic acid retains moisture in the epidermis and maintains the skin's moisturizing function. A decrease in hyaluronic acid is one of the causes of skin dryness. The plasma levels of hyaluronic acid did not change in both males and females; however, after long-term administration of *B. coagulans*, the levels in the skin increased (Figure 2A–D). To investigate the factors that cause skin dryness, we measured cytokine levels, which move in parallel with skin dryness [31,32]. The levels of IL-6 and TNF-α, which are inflammatory cytokines, decreased after long-term *B. coagulans* administration in both males and females (Figure 2E–H). Conversely, the levels of anti-inflammatory cytokines, IL-10, and TGF-β were increased after long-term administration of *B. coagulans* (Figure 2I–L).

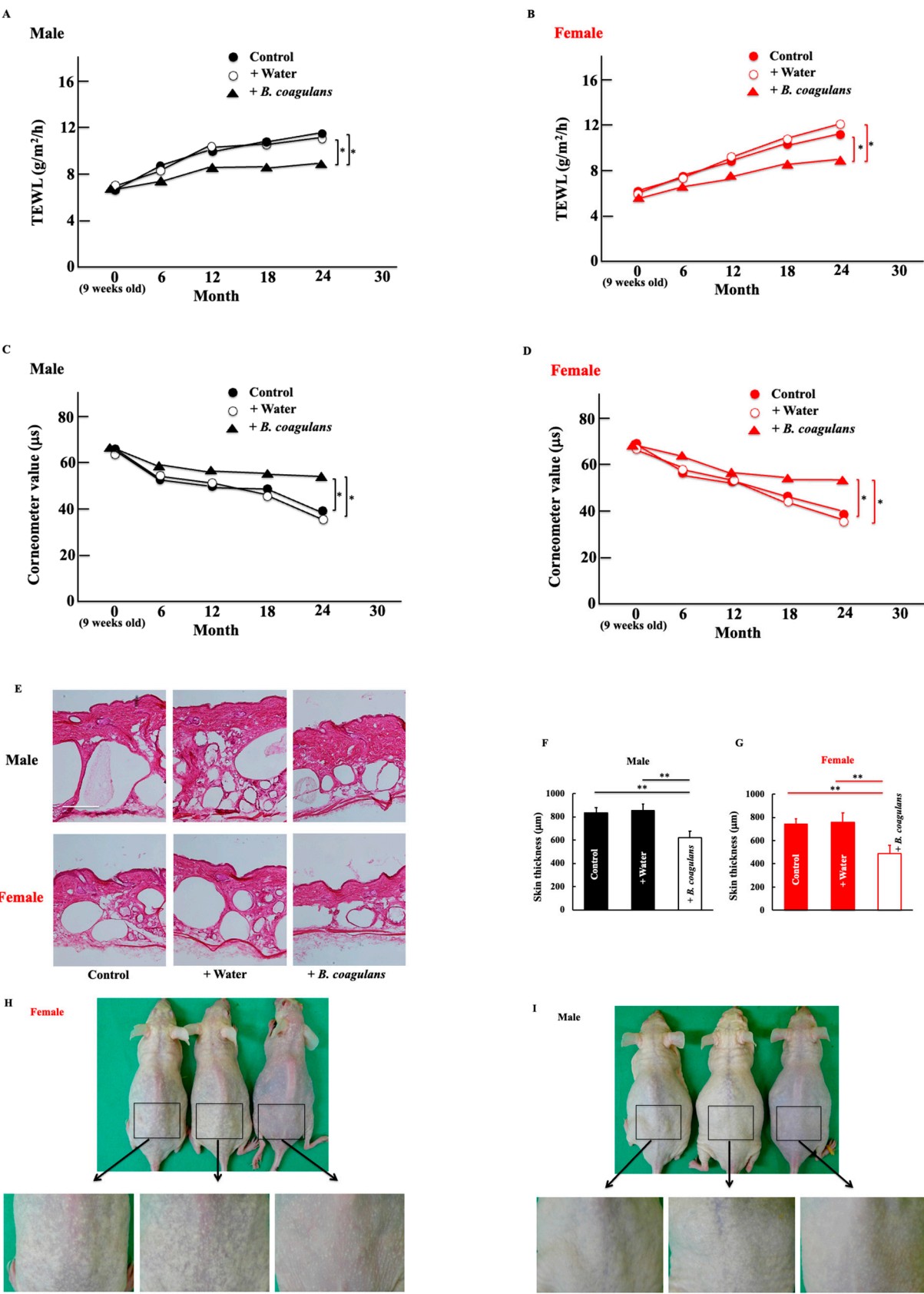

**Figure 1.** *Cont*.

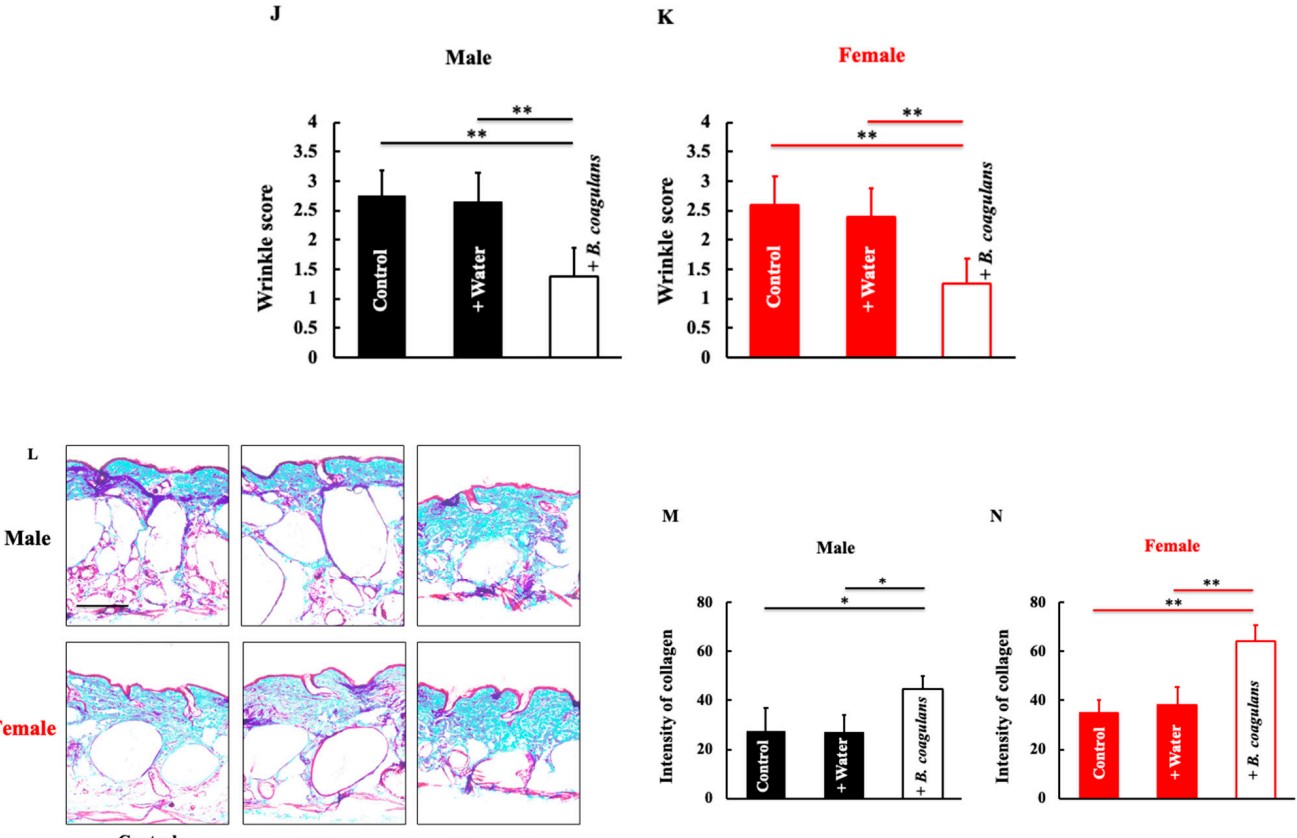

**Figure 1.** Effect of *Bacillus coagulans* administration on natural skin aging on the mouse dorsal skin. At the end of the study, we measured transepidermal water loss (TEWL) (**A**,**B**), corneometer values (**C**,**D**), skin thickness (**E**–**G**), wrinkle scores (**H**–**K**), and the expression of collagen (**L**–**N**) in the dorsal skin of male and female hairless mice. Hematoxylin–eosin staining (**E**) and Masson trichrome staining (**L**). Intensity was calculated from five random visual fields with a constant area using the ImageJ software. Black and red columns correspond to male and female mice, respectively. Scale bar = 100 μm. The values are presented as mean ± standard deviation (SD) of five animals. * *p* < 0.05, ** *p* < 0.01.

### 3.3. Effects of B. coagulans Administration on the Expression of Mast Cells in Aging Mice

The expression of mast cells, correlated with inflammatory cytokines, was measured. Mast cells are recognized as multifunctional effector immune cells [33] and are associated with various pathological conditions such as fibrotic diseases [34] and chronic inflammation [35]. Mast cells contribute to the development of acute and chronic inflammatory responses by releasing preformed and newly synthesized inflammatory mediators [36]. In general, inflammatory mediators and cytokines are significantly increased in elderly patients [37]. Mast cells contribute to the secretion of inflammatory mediators and cytokines during this aging process [38,39]. The expression of mast cells in the skin was reduced by long-term administration of *B. coagulans* in both male and female mice (Figure 3A–C).

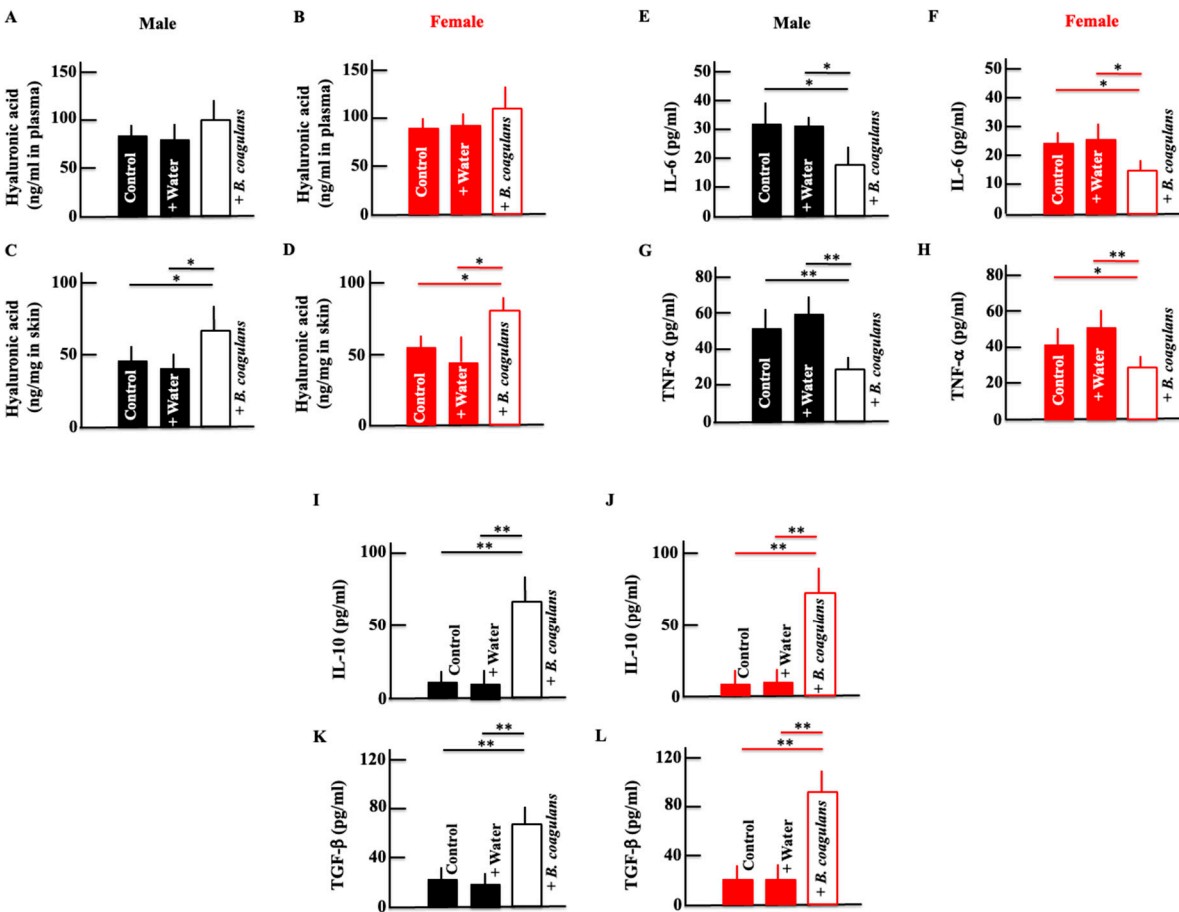

**Figure 2.** Effect of *B. coagulans* administration on the plasma levels of hyaluronic acid (**A**,**B**), IL-6 (**E**,**F**), TNF-α (**G**,**H**), IL-10 (**I**,**J**), and TGF-β (**K**,**L**) and the skin levels of hyaluronic acid (**C**,**D**) in male and female mice. Black and red columns correspond to male and female mice, respectively. The values are presented as mean ± SD of five animals. * $p < 0.05$, ** $p < 0.01$.

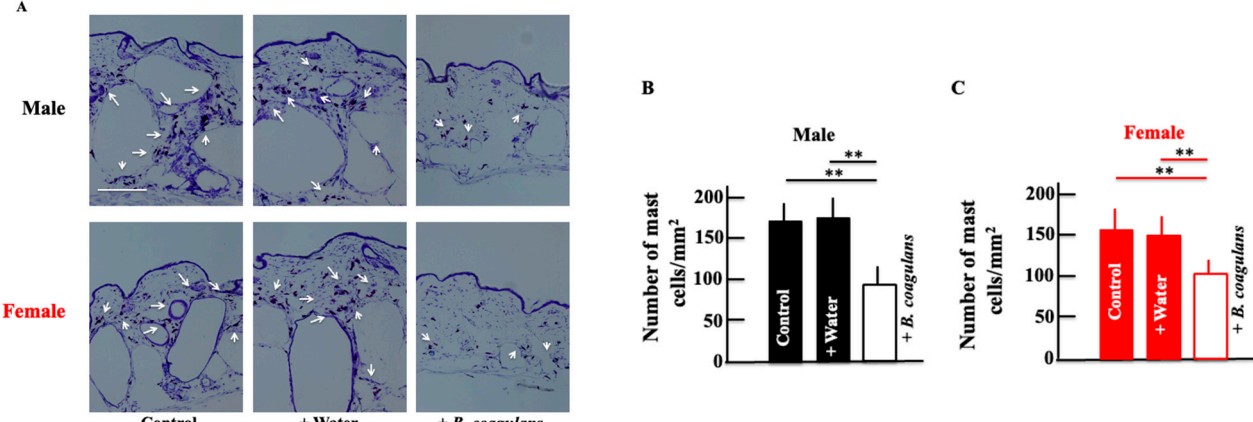

**Figure 3.** Effect of *B. coagulans* administration on the expression of mast cells in the dorsal skin of male (**A**,**B**) and female (**A**,**C**) mice. Skin sections were stained using toluidine blue. Black and red columns correspond to male and female mice, respectively. The values are presented as mean ± SD of five animals. Scale bar = 100 μm. ** $p < 0.01$.

### 3.4. Effects of B. coagulans Administration on Macrophage Levels in Aging Mice

We investigated the macrophages involved in cytokine release. The expression of macrophages themselves (Iba1) did not change in both males and females after long-

term *B. coagulans* administration (Figure 4A,D). However, in both sexes, the M1-type macrophage (inflammatory) marker, CCR7, decreased after long-term *B. coagulans* administration (Figure 4B,E) and the M2-type macrophage (anti-inflammatory) marker, CD163, increased after long-term *B. coagulans* administration (Figure 4C,F).

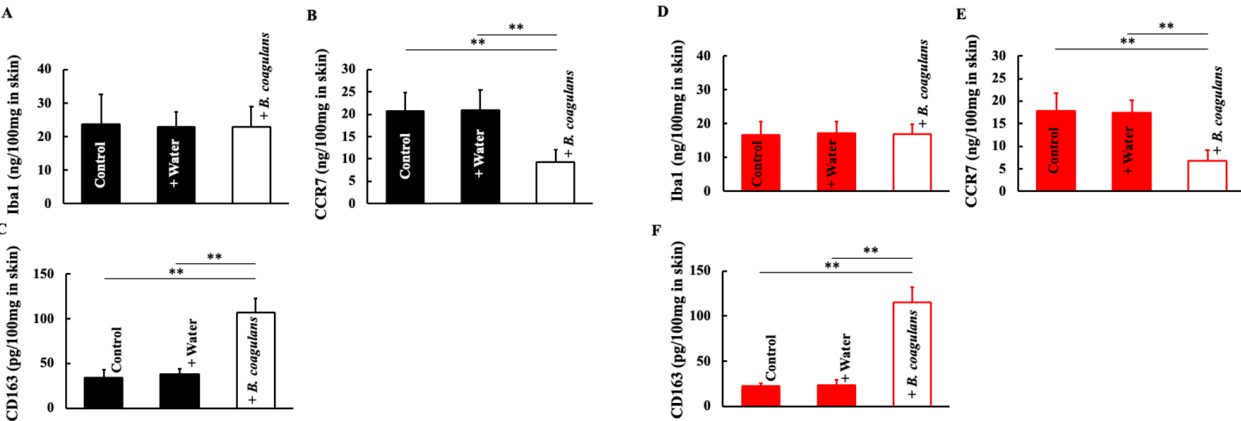

**Figure 4.** Effect of *B. coagulans* administration on macrophage differentiation. At the end of the study, we measured the levels of Iba1 (**A**,**D**), CCR7 (**B**,**E**), and CD163 (**C**,**F**) in the dorsal skin of male and female mice. Black and red columns correspond to male and female mice, respectively. The values are presented as mean ± SD of five animals. ** $p < 0.01$.

### 3.5. Effects of B. coagulans Administration on the Gut Microbiota Profile in Aging Mice

Stacked bar graphs show the genus-level composition of the gut microbiota using microbiome analysis. Lactobacillus was the most dominant fungal genus in the mean values of all groups (Figure 5A). Further, based on the relative abundance ratio of each bacterial genus, we searched for intestinal bacterial genera that differed significantly between the control group and the *B. coagulans* intake group (Figure 5B). As for the *Enterobacter* genera, those with an average relative abundance ratio of 0.001 or more were analyzed. As a result, statistically significant differences were detected for five fungal genera by profiling the microbiome.

### 3.6. Effects of B. coagulans Administration on Metabolites in Aging Mice

To evaluate the effects of *B. coagulans* intake on metabolites, comprehensive measurements of fecal metabolites were performed. This assay detected 352 types of metabolites. Of these, 89 qualified as absolute quantifiable metabolites. The number of moles per gram of stool dry weight for these 89 metabolites is shown in stacked bar graphs (Figure 6A). Short-chain fatty acids such as propionic, butyric, and lactic acid were abundant in the feces of female mice. We further compared fecal metabolites comprehensively among the groups and screened for metabolites that were significantly different between the control and the *B. coagulans* intake group (Figure 6B). As a result, statistically significant differences were detected for 18 metabolites.

**A**

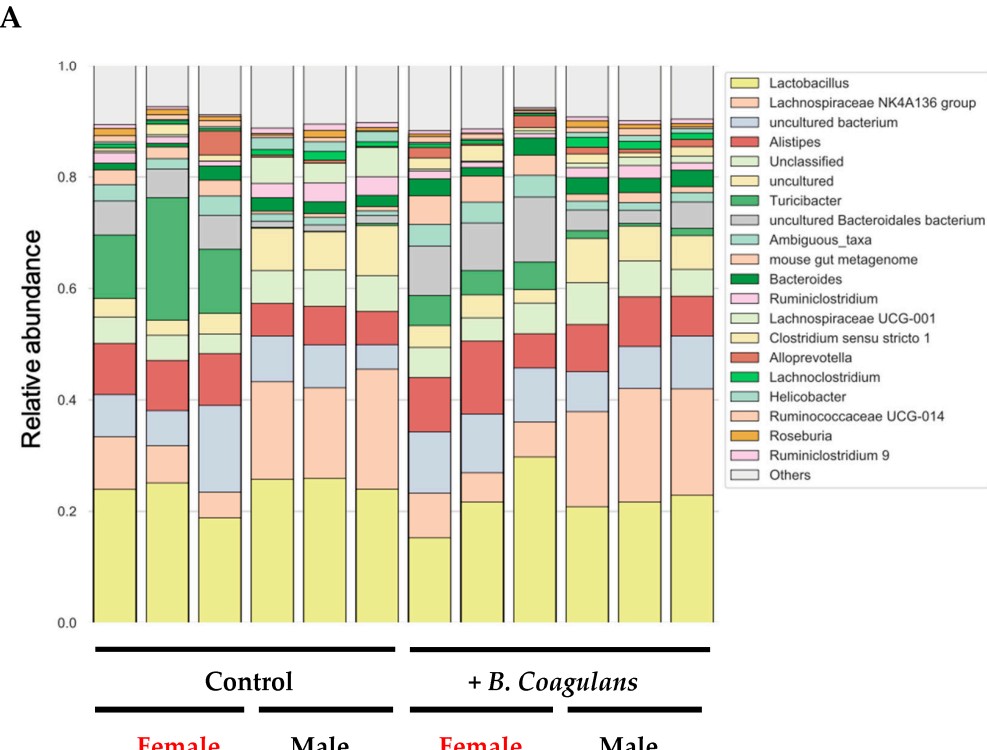

**B**

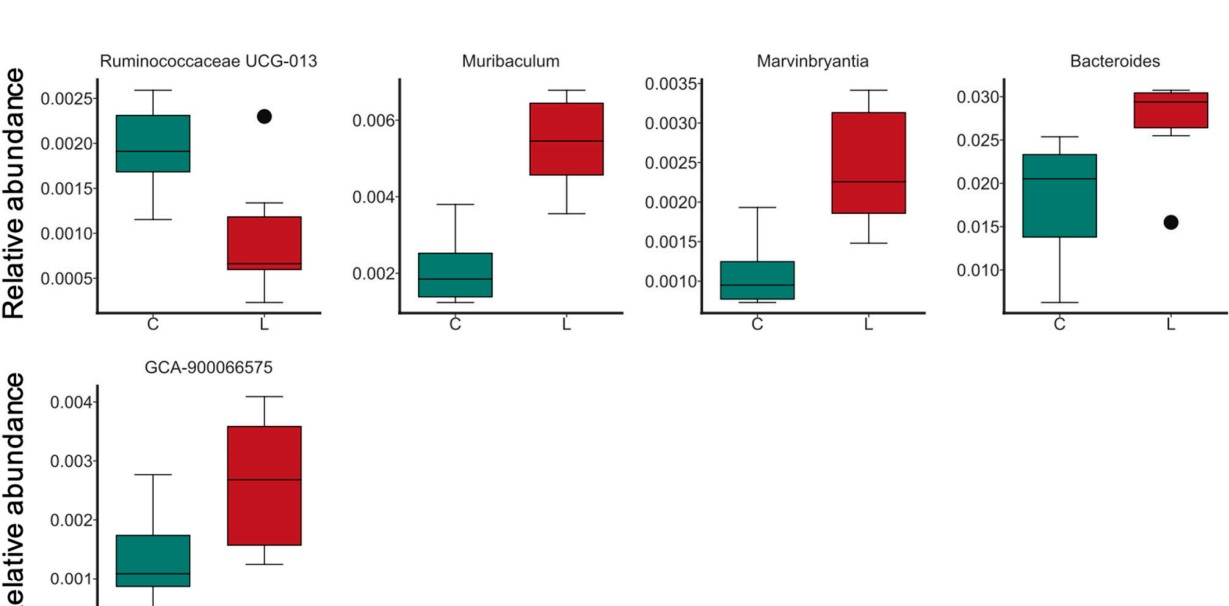

**Figure 5.** Effect of *B. coagulans* administration on gut microbiota profile and enterobacterial genera with significant differences detected in relative abundance ratios. Stacked bar graphs show the relative abundance of each bacterial genus in the intestinal flora (**A**). The 20 genera with the highest average relative abundance of all samples were color-coded, and the other bacterial genera were labeled as "Others". Mouse numbers are indicated below the stacked bar graphs. We compared the relative abundance ratios of enterobacteria between the control group and the *B. coagulans* intake group (Wilcoxon rank sum test) (**B**). In addition, the bacterial family in (**B**) is included in "Others" in (**A**). (**B**) Shows the results for all mice including male and female. C, control group; L, *B. coagulans* intake group. The values are presented as mean ± SD of three animals. Significant differences were observed between C and L in all parameters (*p* < 0.05).

**A**

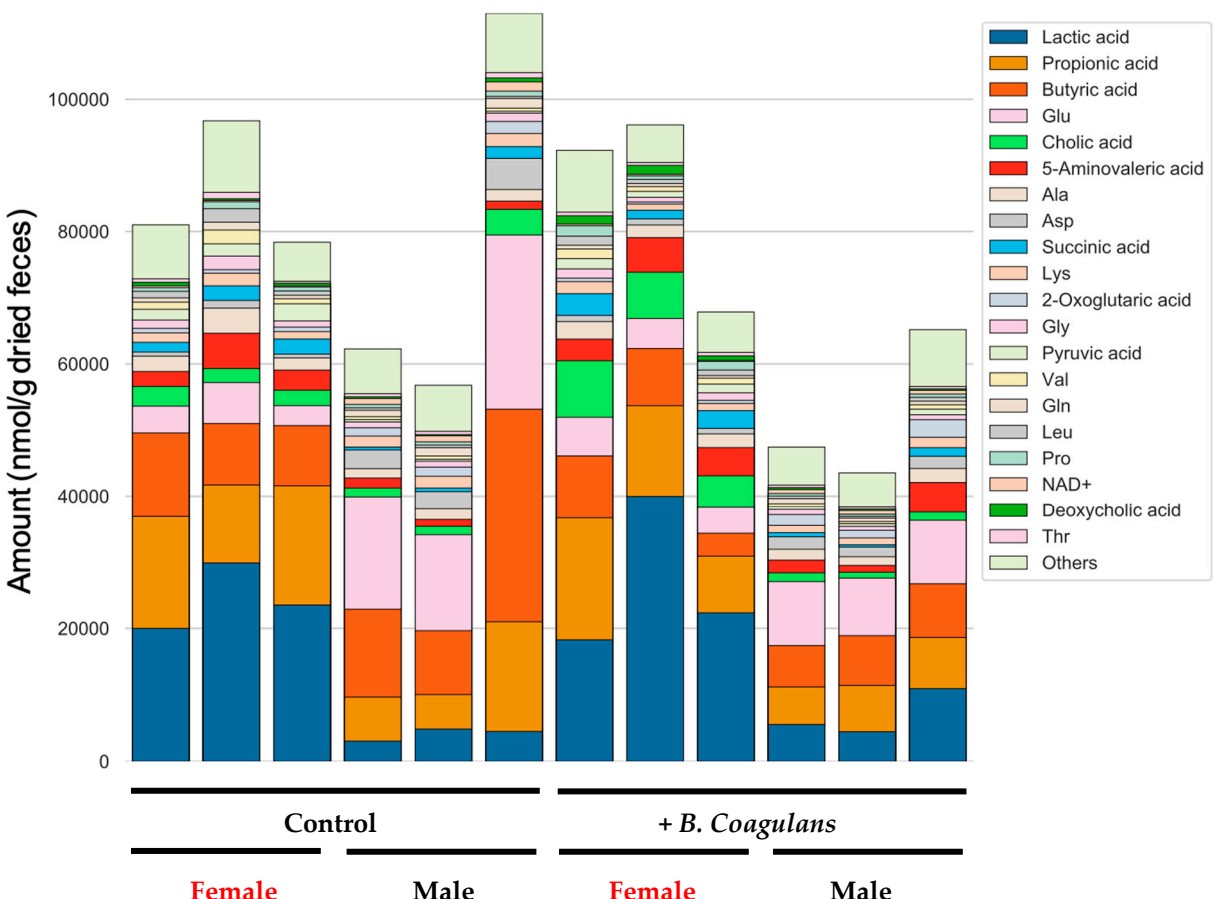

**Figure 6.** *Cont.*

B

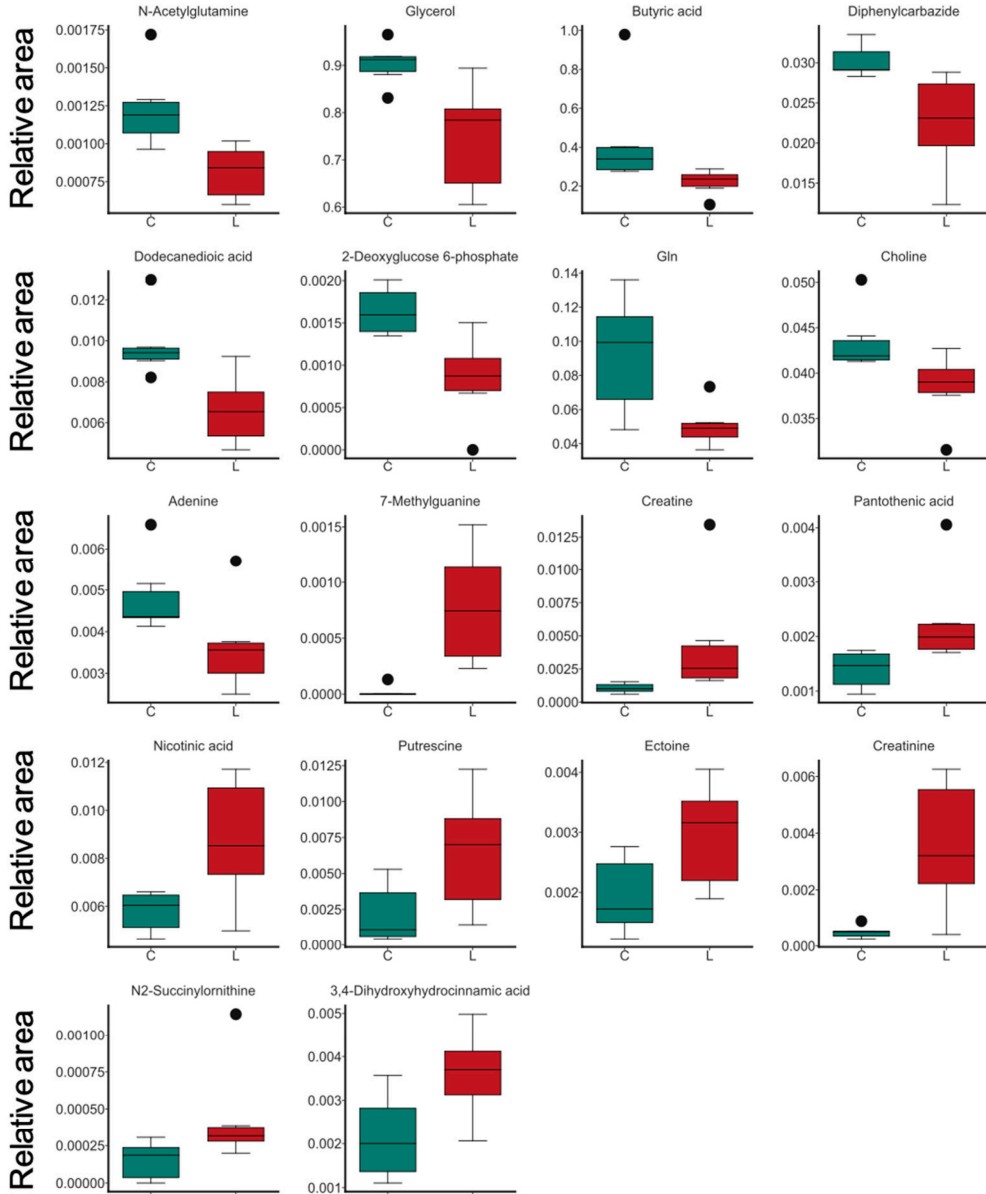

**Figure 6.** Effect of *B. coagulans* administration on the fecal metabolite profile and metabolites with significant differences in relative area ratios. Of the fecal metabolites identified in this study, the quantified 89 types of metabolites are shown in stacked bar graphs (**A**). The 20 metabolites with the highest average amount in all samples were color-coded, and the other metabolites were labeled "Others". The relative area ratios of the control group and the lactic acid bacteria intake group were compared between the two groups (Wilcoxon's rank sum test), and the relative area ratios of the metabolites showing significant differences are shown in boxplots (**B**). (**B**) shows the results for all mice including male and female. C, control group; L, *B. coagulans*. The values are presented as mean ± SD of three animals. Significant differences were observed between C and L in all parameters ($p < 0.05$).

## 4. Discussion

This study demonstrated that the long-term administration of *B. coagulans* reduced TEWL and increased skin moisture retention in spontaneously aging mice of both sexes, indicating an improvement in skin dryness. It also increased the levels of collagen, hyaluronic

acid, type 2 macrophages, and anti-inflammatory cytokines and conversely decreased the levels of inflammatory cytokines and type 1 macrophages. Furthermore, significant differences were observed in many genera of intestinal bacteria and their metabolites between the treatment group and the *B. coagulans* administration group. However, in the results of this study, there was no significant correlation between sex and bacterial genera, whereas significant differences were detected between the control and the *B. coagulans* intake group.

In this study, we investigated the effects of intestinal bacteria and their metabolites on natural skin aging. Regarding intestinal bacteria, there was a significant difference in the relative abundance ratio of five bacterial families between the control and the *B. coagulans* administered group. A significant difference was observed in 18 kinds of metabolites in the presence or absence of *B. coagulans* administration, with no significant correlation with sex. Among these, we further emphasized the skin-related metabolites showing significant differences. Butyric acid can suppress colitis by promoting the differential induction of regulatory T cells, which suppress immune responses [40]. However, in this study, butyric acid levels decreased in the *B. coagulans* administration group, and it was not beneficial in terms of anti-inflammatory effects.

Further, a decrease in choline levels was noted. Choline is a major component of cell membranes and is metabolized to trimethylamine by the intestinal flora. Choline and trimethylamine suppress the expression of genes involved in intestinal epithelial barrier function in the intestines of older adults [41]. Therefore, intestinal permeability may be suppressed owing to a decrease in choline after *B. coagulans* administration.

An increase in pantothenic acid levels was also observed. Human clinical trials suggest that pantothenic acid improves epithelial barrier function by promoting keratinocyte proliferation and fibroblast differentiation [42]. Pantothenic acid has a synthetic pathway gene in intestinal bacteria [43]. Therefore, it is possible that pantothenic acid produced by the intestinal flora was altered by *B. coagulans* inoculation, which improved the skin epithelial barrier function.

In addition, an increase in the levels of nicotinic acid (a form of vitamin $B_3$), a ligand for several receptors (GPR109a) including immune cells, was also observed. Vitamin $B_3$-GPR109a signaling induces the differentiation of regulatory T cells and suppresses colitis in a G-protein-coupled receptor (GPR)109a-dependent manner [44]. Vitamin $B_3$ is produced by a wide range of intestinal bacteria [43]. The significant increase in nicotinic acid in the *B. coagulans* intake group suggests that *B. coagulans* induces the differentiation of regulatory T cells, resulting in an increase in IL-10 and TGF-b, thus exhibiting an inhibitory effect on colitis.

Furthermore, the levels of putrescine, which is a type of polyamine, were found to increase in this study. Putrescine suppresses intestinal permeability [45] and induces anti-inflammatory macrophages (M2 macrophages) which produce anti-inflammatory proteins such as IL-10 [46]. *B. coagulans* ingestion presumably suppressed intestinal permeability through an increase in putrescine levels and suppressed colonic inflammation through an increase in anti-inflammatory macrophages. In addition to these, *B. coagulans* administration increased creatine, creatinine, and ectoine levels and decreased N-acetylglutamine levels. However, the role of these metabolites is still unknown.

Based on these analyses, Figure 7 shows the correlation among skin conditions, enterobacteria, metabolites, and inflammation-related substances examined in this study. These analyses revealed four skin function improvement mechanisms through intestinal bacteria induced by *B. coagulans* administration.

| Bacteria/Metabolites | TEWL Increase or decrease | Skin hydration | Hyaluronic acid (blood) | Hyaluronic acid (skin) | IL-6 | TNF-α | IL-10 | TGF-β | Mast cell count | CCR7 | CD163 |
|---|---|---|---|---|---|---|---|---|---|---|---|
| Ruminococcaceae UCG-013 | ↓ | ○ | | | ○ | ○ | | ○ | | ○ | |
| *Muribaculum* | ↑ | ○ | ○ | ○ | ○ | | ○ | ○ | | ○ | ○ | ○ |
| *Marvinbryantia* | ↑ | | ○ | ○ | | ○ | ○ | ○ | | ○ | ○ | ○ |
| *Bacteroides* | ↑ | | ○ | | ○ | | | ○ | | ○ | |
| GCA-900066575 | ↑ | | ○ | ○ | | | ○ | | | ○ | |
| N-Acetylglutamine | ↓ | ○ | ○ | ○ | ○ | | ○ | ○ | ○ | | ○ | ○ |
| Glycerol | ↓ | | ○ | ○ | ○ | ○ | ○ | ○ | | ○ | ○ | ○ |
| Butyric acid | ↓ | ○ | ○ | ○ | ○ | | ○ | ○ | ○ | ○ | ○ | ○ |
| Diphenylcarbazide | ↓ | ○ | ○ | ○ | ○ | | ○ | ○ | ○ | ○ | ○ | ○ |
| Dodecanedioic acid | ↓ | ○ | ○ | ○ | | | ○ | | ○ | ○ | ○ | ○ |
| 2-Deoxyglucose 6-phosphate | ↓ | ○ | ○ | ○ | ○ | ○ | | | | | ○ | |
| Gln | ↓ | ○ | ○ | ○ | ○ | | | | ○ | | ○ | |
| Choline | ↓ | | ○ | ○ | ○ | | | ○ | | | ○ | ○ |
| Adenine | ↓ | | | | ○ | ○ | | ○ | | ○ | | ○ |
| 7-Methylguanine | ↑ | ○ | ○ | ○ | ○ | ○ | ○ | ○ | ○ | ○ | ○ | ○ |
| Creatine | ↑ | ○ | ○ | ○ | ○ | ○ | ○ | ○ | ○ | ○ | ○ | ○ |
| Pantothenic acid | ↑ | | ○ | | ○ | ○ | | ○ | ○ | ○ | ○ | ○ |
| Nicotinic acid | ↑ | ○ | ○ | | ○ | ○ | ○ | | | | | |
| Putrescine | ↑ | ○ | ○ | ○ | ○ | ○ | ○ | | | ○ | ○ | |
| Ectoine | ↑ | ○ | ○ | | ○ | | | ○ | | ○ | | |
| Creatinine | ↑ | ○ | ○ | ○ | | | ○ | ○ | | ○ | ○ | |
| N2-Succinylornithine | ↑ | ○ | ○ | | | ○ | ○ | | | ○ | | |
| 3,4-Dihydroxyhydrocinnamic acid | ↑ | ○ | | ○ | | ○ | ○ | | ○ | | | |

**Figure 7.** Correlations between bacteria/metabolites and metadata showing significant differences between the control and the *B. coagulans* intake group. A two-group comparison (Wilcoxon rank sum test) was performed between the control group and the *B. coagulans* intake group, and the correlation results with each metadata are shown for bacteria and metabolites for which a significant difference ($p < 0.05$) was detected. No correlation was found between Iba1 and sex. Therefore, this column was omitted. The legends for each column are as follows: "Increase or decrease" column—↑: high in the *B. coagulans* intake group (pink background color), ↓: low in the *B. coagulans* intake group (blue background color); Other columns—○. A significant correlation was observed (pink background color; $p < 0.05$ in the Spearman correlation coefficient uncorrelation test).

1. Increased absorption of hyaluronic acid from the intestine

Hyaluronic acid is present abundantly in the skin and prevents skin dryness due to its water retention capacity. Orally ingested hyaluronic acid is degraded and absorbed by intestinal bacteria [47]. Bacteria belonging to the genera *Bacteroides* and *Lactobacillus* degrade hyaluronic acid [48]. In this study, the ingestion of *B. coagulans* significantly increased the abundance of the genus *Bacteroides* and hyaluronic acid levels, and a significant correlation was observed between the proportion of *Bacteroides* and hyaluronic acid levels in the skin. Thus, various intestinal bacteria, primarily *Bacteroides*, may increase the decomposition and absorption of hyaluronic acid in the intestine, resulting in an increase in hyaluronic acid levels in the skin.

2. Induction of IL-10-producing cells

IL-10 is an anti-inflammatory cytokine and is essential for improving skin health [49]. In this analysis, a significant increase in the proportion of *Bacteroides* spp. was observed with *B. coagulans* intake. The genus *Bacteroides* may promote IL-10 production; however, identification at the species level is difficult. Nicotinic acid-GPR109a signaling induces differentiation of regulatory T cells and IL-10-producing T cells [44]. Therefore, increased nicotinic acid levels may increase IL-10 production. In addition, putrescine induces IL-10-producing anti-inflammatory macrophages (M2 macrophages) [46]. In this study, no significant correlation was observed between IL-10, nicotinic acid, and putrescine levels;

however, a significant correlation was observed between IL-10 levels and bacterial abundance of the genus *Bacteroides*. Therefore, IL-10 levels may have increased due to an increase in the abundance of the genus *Bacteroides*. On the other hand, it is thought that long-term *Bacillus coagulans* administration shifts macrophages to anti-inflammatory macrophages (M2 macrophages) in the skin, and conversely decreases inflammatory macrophages (M1 macrophages), resulting in a decrease in the secretion of IL-6 and TNF-a. However, the details are not known.

3. Suppression of intestinal permeability

Intestinal bacteria and food antigens are present in the intestinal lumen. With an increase in intestinal permeability, these substances flow into the body and induce immune and inflammatory reactions. Furthermore, increased intestinal permeability increases blood levels of metabolites, such as phenol and para-cresol, which eventually reduce skin function [50]. In this study, a significant increase in putrescine levels and a significant decrease in choline levels were observed in the *Muribaculum* spp. *Muribaculum* bacteria; putrescine suppresses intestinal permeability and choline enhances intestinal permeability [27,40,43]. Thus, skin symptoms may improve by suppressing intestinal permeability and reducing metabolites that impair skin function.

4. Increased pantothenic acid levels

Ingestion of pantothenic acid improves epithelial barrier function [41]. In this study, the administration of *B. coagulans* also increased fecal pantothenic acid levels, suggesting that an increase in intestinal pantothenic acid levels may improve epithelial barrier function.

This study has a few limitations that warrant discussion. This study does not elaborate on the analysis of intestinal bacterial species involved in improving skin aging. Further, although differential metabolites were identified, their potential role in skin dryness is yet to be established. Therefore, it is necessary to conduct more detailed tests and examine the involvement of intestinal flora, metabolites, and inflammatory substances. Furthermore, the gut microbiota of mice and humans may differ; hence, confirmation through human clinical trials is needed. In addition, although there is a sex-based difference in intestinal bacteria, this study's findings did not find a sex-dependent difference in skin properties; therefore, it is necessary to investigate this factor.

## 5. Conclusions

This study revealed that the administration of *B. coagulans* improved the natural skin aging in mice. This enhancement might be induced by the interaction of alterations in intestinal flora, metabolites, or inflammatory substances.

**Author Contributions:** K.H., S.K., K.T., Y.I. and H.H. wrote the article and designed the research. K.H., S.K. and D.S. analyzed and interpreted the data. All authors have read and agreed to the published version of the manuscript.

**Funding:** This study was supported by a JSPS KAKENHI grant (number: 23K06074).

**Institutional Review Board Statement:** All experimental procedures described in the present study were conducted in accordance with the recommendations of the Guide for the Care and Use of Laboratory Animals of the Suzuka University of Medical Science (Approval number: 34). All surgeries were performed under pentobarbital anesthesia and efforts were made to minimize animal suffering.

**Informed Consent Statement:** Not applicable.

**Data Availability Statement:** Data are available within the article.

**Acknowledgments:** Iizuka and Hamano contributed the essential reagents and tools. We would like to thank Vikas Narang for English language editing. The *B. coagulans* samples used in this study were provided by Daiichi Sankyo Healthcare Co., Ltd. (Tokyo, Japan).

**Conflicts of Interest:** Authors S. Kubo, K. Tsuji, D. Sugiyama, Y. Iizuka, and H. Hamano were employed by the company Daiichi Sankyo Healthcare Co., Ltd. The authors declare that this study received funding from Daiichi Sankyo Healthcare Co., Ltd. The funder was not involved in the study design, collection, analysis, interpretation of data, the writing of this article, or the decision to submit it for publication.

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
