# Peer review of "The Effect of Bacillus coagulans Induced Interactions among Intestinal Bacteria, Metabolites, and Inflammatory Molecules in Improving Natural Skin Aging"

_dermatopathology, doi:10.3390/dermatopathology10040037_

Round 1

Reviewer 1 Report

General Comments:

1.       Please provide more background/rationale for choosing to administer B. coagulans, specifically, for this study.

2.       When reporting results from this study, the authors should be careful to accurately compare mice that were administered B. coagulans for the two-year study duration to the control group(s). For instance, it is inaccurate to state that “wrinkles on the dorsal skin improved after B. coagulans administration” (line 163) or that there is “an increased amount of collagen fibers after B. coagulans treatment (line 165-6). These statements imply that mice were aged, parameters were measured, and a short-term B. coagulans treatment “improved” or reversed signs of aging which is not true. It is presumed that the B. coagulans group never formed skin wrinkles or decreased collagen levels in the first place due to the specified long-term treatment. These types of statements continue throughout (e.g., lines 182, 186). Please correct the interpretation of results to accurately reflect the study design.

3.       In section 3.2, please provide rationale for assessing cytokines as a measure of skin dryness.

4.       In section 3.3, please provide more background information (with associated references) that describes the relationship between mast cells and skin aging.

Figure Comments:

5.       Figure 1. Panels H and I. Increasing resolution of the photographs would be helpful. It is difficult to visualize the wrinkles.

6.       Figure 1. Panels L-N. Please clarify in legend the staining method (Trichrome?) and brief description of quantitation (Does the “intensity of collagen” mean the amount?)

7.       Figures 5 and 6. Enlarge and increase resolution. These figures are very difficult to read.

8.       Several bacterial genera are represented by bar graphs in Figure 5B (e.g., Muribaculum, Marvinbryantia, GCA-900….) that are not present in the stacked graphs in Figure 5A. Are these represented in “Others”? Please clarify.

9.       Are sex-dependent differences considered in Figures 5B and 6B? It is mentioned in the discussion that there is no correlation between sex and metabolites (lines 265-7), but that “there is a sex-based difference in intestinal bacteria” (lines 346-7) which is not clear from Figure 5B. Are the data in Figures 5B and 6B from male and female mice combined? Please clarify.

Minor Comments:

10.   Lines 156-7. Moisture retention did not increase with age, as stated. TEWL increased with age. Please correct.

11.   Line 181. “hyaluronic acid, which causes skin dryness”. Hyaluronic acid is associated with skin moisture, not dryness. Please correct or clarify this meaning.

12.   Line 222. It is stated that “differences were detected for five fungal genera.” It is assumed that bacterial genera would be determined through profiling the microbiome. Please correct or clarify.

13.   An expanded description of results from Figure 5B may be helpful.

Author Response

Response to the Reviewers’ comments

To Reviewer 1

We would like to thank the reviewer for appreciation of our work.

Thank you very much for your valuable comments on this paper.

General Comments:

  1. Thank you for your comment. We have included a more detailed background/rationale using B. coagulans in the introduction.

(Introduction: lines 68-73):

Furthermore, we demonstrated that B. coagulans administration had an ameliorating effect on AOM+DSS-induced colon cancer [22]. It was revealed that B. coagulans induces an increase in transforming growth factor-β (TGF-β) and regulates the immune system. Chronic inflammation is thought to be one of the causes of aging, and inflammatory cytokines have been shown to play an important role [23]. B. coagulans was predicted to affect skin aging through its involvement with the immune system.

[22] Hiramoto, K.; Kubo, S.; Tsuji, K.; Sugiyama, D.; Iizuk, Y.; Hamana, H. Bacillus. coagulans (species of lactic acid-forming Bacillus bacteria) ameliorates azoxymethane and dextran sodium sulfate-induced colon cancer in mice. J. Funct. Foods, 2023, 100, 10406.

[23] Yamaguchi, Y. Periostin in skin tissue and skin-related disease. Allergol. Int. 2014, 63(2), 161-170.

  1. Thank you for your comment. As the reviewer pointed out, short-term administration did not improve skin aging. B. coagulans suppresses the formation of wrinkles in the skin and the decline of collagen each time it is administered over a long period of time, and it is thought that this ultimately resulted in a significant difference. We have rewritten the expression of the results.

  (Results: lines 176-177):

  Moreover, after long-term administration of B. coagulans, dorsal skin wrinkles were improved.

   (Results: lines 178-179):

   Histological examination of the skin of male and female mice showed increased collagen fiber content after long-term B. coagulans administration.

   (Results: lines 199-200):

   However, after long-term administration of B. coagulans, the levels in the skin Increased.

   (Results: line 203):

   long-term administration

   (Results: lines 204-205):

   The levels of anti-inflammatory cytokines, IL-10, and TGF-β were increased after long-term administration of B. coagulans.

   (Results: lines 221-222):

   The expression of mast cells in the skin was reduced by long-term administration of B. coagulans.

   (Results: line 231):

   long-term administration

(Results: line 233):

   long-term administration

   (Results: line 235):

   long-term administration

  1. Thank you for your comment.

Since skin dryness and cytokines are known to work in parallel, we measured representative cytokines.

(Results: lines 201-202):

To investigate the factors that cause skin dryness, we measured cytokine levels, which move in parallel with skin dryness [31,32].

[31] Hiramoto, K.; Sugiyama, D.; Takahashi, Y.; Mafune, E. The amelioration effect of tranexamic acid in wrinkles induced by skin dryness. Biomed. Pharmacother. 2016, 80, 16-22.

[32] Tominaga, M.; Takamori, K. Peripheral itch sensitization in atopic dermatitis. Allergol. Int. 2022, 71(3), 265-277.

  1. Thank you for your comment. We have added detailed background information explaining the relationship between mast cells and skin aging in Section 3.3.

   (Results: lines 215-221):

  Mast cells are recognized as multifunctional effector immune cells [33] and are associated with various pathological conditions such as fibrotic diseases [34] and chronic inflammation [35]. Mast cells contribute to the development of acute and chronic inflammatory responses by releasing preformed and newly synthesized inflammatory mediators [36]. In general, inflammatory mediators and cytokines are significantly increased in elderly patients [37]. Mast cells contribute to the secretion of inflammatory mediators and cytokines during this aging process [38,39].

   [33] Elbasiony, E.; Mittal, S.K.; Foulsham, W.; Cho, W.K.; Chauhan, S.K. Epithelium derived IL-33 activates mast cells to initiate neutrophil recruitment following corneal injury. Ocul. Surf. 2020, 18, 633-640.

   [34] Veerappan, A.; O’Connor, N.J.; Brazin, J.; Reid, A.C.; Jung, A.; McGee, D.; Summers, B.; Branch-Elliman, D.; Stiles, B.; Worgall, S.; et al. Mast cells: a pivotal role in pulmonary fibrosis. DNA Cell Biol. 2013, 32(4), 206-218.

   [35] Metz, M.; Grimbaldeston, M.A.; Nakae, S.; Piliponsky, A.M.; Tsai, M.; Galli, S.J. Mast cells in the promotion and limitation of chronic inflammation. Immunol. Rev. 2007, 217, 304-328.

   [36] Krystel-Whittemore, M.; Dileepan, K.N.; Wood, J.G. Mast cell: a multi-functional master cell. Front Immunol.2016, 6, 620.

   [37] Mendeley, K.S.; Pedersen, M.; Bruunsgaard, H. Inflammatory mediators in the elderly. Exp. Gerontol. 2004, 39(5), 687-699.

   [38] Elbasony, E.; Cho, W.J.; Singh, A.; Mittal, S.K.; Zoukhri, D.; Chauhan, S.K. Increased activity of lacrimal gland mast cells are associated with corneal epitheliopathy in aged mice. NPJ Aging, 2023, 9(1), 2.

   [39] Lin, S.; Huang, H.; Ling, M.; Zhang, C.; Yang, F.: Fan, Y. Development and validation of a novel diagnostic model for musculoskeletal aging (sarcopenia) based of cuproptosis-related genes associated with immunity. Am. J. Transl. Res. 2022, 14(12), 8523-8538.

Figure Comments:

  1. Thank you for your comment. I have increased the resolution of the wrinkles photo, but I couldn't see them clearly. From now on, we will use a more sensitive camera to take pictures.

  1. Thank you for your comment. We have added a brief description of the staining method and quantification.

   (Figure legend 1: lines 190-192):

   Hematoxylin-eosin staining (E) and Masson trichrome staining (L). Intensity was. calculated from five random visual fields with constant area using the ImageJ software.

  1. Thank you for your comment. We increased the resolution of Figures 5 and 6.

  1. Thank you for your comment. In addition, the bacterial family in 5B is included in others in 5A.

    (Figure legend 5: lines 267-268):

    In addition, the bacterial family in (B) is included in others in (A).

  1. Thank you for your comment. Figures 5B and 6B show the results for all mice, including male and female.

  (Figure legend 5: line 268):

  (B) shows the results for all mice including male and female.

(Figure legend 6: lines 334-335):

  (B) shows the results for all mice including male and female.

Minor Comments:

  1. Thank you for your comment. Moisturizing capacity decreases with age, and water transpiration increases. We rewrote it.

    (Results: lines 169-171):

    Water retention, an indicator of skin dryness, decreased with age in both male and. female mice. Additionally, water transpiration increased with age in both male and female mice. This effect was significantly suppressed by long-term administration of B. coagulans (Figure 1A and B).

  1. Thank you for your comment. Hyaluronic acid retains moisture in the epidermis and maintains the skin's moisturizing function. A decrease in hyaluronic acid is one of the causes of skin dryness.

   (Results: lines 197-198):

   Hyaluronic acid retains moisture in the epidermis and maintains the skin's. moisturizing function. A decrease in hyaluronic acid is one of the causes of skin dryness.

  1. Thank you for your comment. Bacterial genera have been determined through microbiome profiling.

   (Results: lines 250):

   As a result, statistically significant differences were detected for five fungal genera. through profiling the microbiome.

  1. Thank you for your comment. I am very sorry, but it is difficult to explain about Figure 5B, so I cannot discuss it in detail. I think further consideration is required.

Reviewer 2 Report

This study investigates the effects of Bacillus Coagulans administration on Natural Skin Aging and its influence on the interaction between gut bacteria, metabolites, and inflammatory substances.
The study is very interesting, but needs improvement in the following areas.

(1) The methods section of the abstract should describe the detailed methods, but the results are described.

(2) In the introduction, it would be better to describe the previously reported sex difference studies in Natural Skin Aging and the reason why male and female mice were compared in this study.

(3) The authors examine the effect of Bacillus Coagulans administration on Natural Skin Aging. However, there are no comparative data between young and aged mice, and thus it is unclear whether Bacillus Coagulans suppresses the changes caused by Aging.
In this regard, additional data from young mice or reports from other groups may be needed.

(4) The authors claim that the suppressive effect of Bacillus Coagulans administration on Natural Skin Aging is related to IL-10 and TGF-b, but their data are based on plasma concentrations only. Data on the concentration of these anti-inflammatory substances and gene expression in the skin would be more convincing. Or it should be mentioned in the text.

(5) The mechanism by which Bacillus Coagulans administration suppresses the aging-induced increase in IL-6 and TNF-a should be mentioned in the discussion.

(6) The pentobarbital anesthesia used in this animal studies is not recommended for use alone as a general anesthetic in principle in many institutions, and therefore may require ethical confirmation.

A native English language check of the text may be necessary.

Author Response

Response to the Reviewers’ comments

To Reviewer 2

We would like to thank the reviewer for appreciation of our work.

(1) Thank you for your comment. We put "Methods" in the wrong place in the summary. I will correct it.

(Abstract)

Methods: This study examined the effects of skin naturalization (particularly skin drying) by ad-ministering a spore-bearing lactic acid bacteria (Bacillus coagulans) in mice for 2 years.

(2) Thank you for your comment. We have added the reason why male and female mice were compared in this study.

(Introduction: lines 41-47)

Furthermore, we have repeatedly investigated the effects of gender differences on natural aging [7,8]. A comparison of males and females showed that females suppressed aging more, and that estradiol was involved in this effect. It has been suggested that estradiol inhibits hyaluronic acid degrading enzyme and MMP-1, thereby preventing the decrease of hyaluronic acid and collagen in the skin and suppressing skin aging. In this way, since there are gender differences in the natural aging of the skin, gender differences were also investigated in this study.

[7] Hiramoto, K.; Orita, K.; Yamate, Y.; Kobayashi, H. Role of Momordica charantia in preventing the natural aging process of skin and sexual organs in mice. Dermatol. Ther. 2022, 33(6), e14243.

[8] Hiramoto, K.; Yamate, Y.; Sugiyama, D.; Matsuda, K.; Iizuka, Y.; Yamaguchi, T. Ameliorative effect of tranexamic acid on physiological skin aging and its sex difference in mice. Arch. Dermatol. Res. 2019, 311(7), 545-553.

(3) Thank you for your comment. As pointed out by reviewer, there is no comparative data between young and old mice. We need data on young mice and would like to consider this. Regarding the condition of the skin, there is data in a previous report comparing it with a young one (9 weeks old).

Hiramoto, K.; Yamate, Y.; Sugiyama, D.; Matsuda, K.; Iizuka, Y.; Yamaguchi, T. Ameliorative effect of tranexamic acid on physiological skin aging and its sex difference in mice. Arch. Dermatol. Res. 2019, 311(7), 545-553.

(4) Thank you for your comment. We did not review data regarding the expression of these anti-inflammatory substances in the skin. As the reviewer pointed out, expression data in the skin would be convincing. From next time onwards, we will collect data on gene expression in the skin.

(5) Thank you for your comment. The mechanism by which Bacillus coagulans administration suppresses the aging-induced increase in IL-6 and TNF-α is considered as follows.

(Discussion: lines 411-415)

On the other hand, it is thought that long-term Bacillus Coagulans administration shifts macrophages to anti-inflammatory macrophages (M2 macrophages) in the skin, and conversely decreases inflammatory macrophages (M1 macrophages), resulting in a decrease in the secretion of IL-6 and TNF-α. However, the details are not known.

(6) Thank you for your comment. The use of pentobarbital anesthesia was carried out in strict accordance with the recommendations of the Guide for the Care and Use of Laboratory Animals of Suzuka Medical University.

Thank you for your comment. This paper was corrected by a native English speaker.

Round 2

Reviewer 1 Report

My previous comments have largely been addressed in the revised manuscript. I recommend to accept this manuscript in its present form.

Reviewer 2 Report

This is a very interesting and important area of research.

In this paper, the authors demonstrate the importance of the relationship between intestinal bacteria, their metabolites, and inflammatory molecules in the efficacy of Bacillus Coagulans on Natural Skin Aging.

I believe that further investigation of the mechanism of how these molecules are involved in each other will clarify the background of Natural Skin Aging.